# Sensing and signaling of immunogenic extracellular RNAs restrain group 2 innate lymphoid cell-driven acute lung inflammation and airway hyperresponsiveness

Li She [1,2◦], Hamad H. Alanazi [1◦], Liping Yan[1], Edward G. Brooks [3], Peter H. Dube[1], Yan Xiang [1], Fushun Zhang[1], Yilun Sun[1], Yong Liu[2], Xin Zhang[2], Xiao-Dong Li [1◦]*

1 Department of Microbiology, Immunology and Molecular Genetics, University of Texas Health Science Center at San Antonio, San Antonio, TX, United States of America, 2 Department of Otolaryngology-Head and Neck Surgery, Xiangya Hospital, Central South University, Changsha, Hunan, China, 3 Division of Immunology and Infectious Disease, Long School of Medicine, University of Texas Health San Antonio, San Antonio, TX, United States of America

◦ These authors contributed equally to this work.

* lix8@uthscsa.edu

## Abstract

Repeated exposures to environmental allergens in susceptible individuals drive the development of type 2 inflammatory conditions such as asthma, which have been traditionally considered to be mainly mediated by Th2 cells. However, emerging evidence suggest that a new innate cell type, group 2 innate lymphoid cells (ILC2), plays a central role in initiating and amplifying a type 2 response, even in the absence of adaptive immunity. At present, the regulatory mechanisms for controlling ILC2 activation remain poorly understood. Here we report that respiratory delivery of immunogenic extracellular RNA (exRNAs) derived from RNA- and DNA-virus infected cells, was able to activate a protective response against acute type 2 lung immunopathology and airway hyperresponsiveness (AHR) induced by IL-33 and a fungal allergen, *A. flavus*, in mice. Mechanistically, we found that the innate immune responses triggered by exRNAs had a potent suppressive effect *in vivo* on the proliferation and function of ILC2 without the involvement of adaptive immunity. We further provided the loss-of-function genetic evidence that the TLR3- and MAVS-mediated signaling axis is essential for the inhibitory effects of exRNAs in mouse lungs. Thus, our results indicate that the host detection of extracellular immunostimulatory RNAs generated during respiratory viral infections have an important function in the regulation of ILC2-driven acute lung inflammation.

## Introduction

Asthma places a huge socioeconomic burden on our society and the number of people with asthma has continued to rise globally in recent decades [1–3]. Despite enormous research efforts, the etiology and molecular mechanism of asthma pathogenesis are still poorly

**Data Availability Statement:** All relevant data are within the paper and its Supporting Information files.

**Funding:** LS is supported by the China Scholarship Council and Hunan Provincial Innovation Foundation for Postgraduate (CX201713068). HHA is supported by the Department of Clinical Laboratory Sciences, College of Applied Medical Sciences, Jouf University, Sakaka, Saudi Arabia. X.-D.L. is supported by the UT Health San Antonio School of Medicine Startup Fund and the Max and Minnei Voelcker Fund.

**Competing interests:** No authors have competing interests.

**Abbreviations:** A. flavus, *Aspergillus flavus*; AHR, airway hyperresponsiveness; dsRNA, double-stranded RNA; exRNAs, extracellular RNAs; IL-33, interleukin-33; PRRs, pattern recognition receptors; P(Poly[I:C]), olyinosinic:polycytidylic acid; Th2, T helper 2 cell; Trif, TIR-domain-containing adapter-inducing interferon-β; ILC2, toll-like receptor 3 (TLR3), type 2 innate lymphoid cells; IFNAR1, (interferon alpha/beta receptor 1).

understood. Environmental allergen exposure triggers aberrant type 2 immune responses in susceptible individuals, but not in healthy individuals. The underlying mechanisms remain largely undefined, but emerging evidence in recent years supports the 'hygiene hypothesis' orginally propoposed by Strachan [4]. Environmental endotoxin has been recently shown to protect against the development of allergy [5, 6], suggesting that humans have evolved protective innate immune pathways to maintain tissue homeostasis by counteracting the development of a polarized type 2 responses through sensing the environment. Allergen recognition through pattern recognition receptors (PRRs) is thought to be the critical step that determines either mucosal homeostasis or the development of inappropriate type 2 inflammation [5, 7–10]. In reality, humans are naturally exposed to environmental allergens that are often in complex forms containing both allergic proteins that are responsible for inducing type 2 inflammatory responses, and microbial substances such as nucleic acids (RNA/DNA), which may act as an immune adjuvant by triggering nucleic acid immunity [11–13].

In the past few years, it has become increasingly appreciated that group 2 innate lymphoid cells (ILC2) play a central role in the initiation and orchestration of type 2 inflammatory diseases such as asthma [14]. ILC2, lacking rearranged antigen receptors, is the innate counterpart of the T helper 2 lymphocyte (Th2) [14–17]. In response to environmental signals, or inducer cytokines such as Interleukin-33 (IL-33), ILC2 can rapidly produce massive amounts of the type 2 cytokines IL-5 and IL-13 [18–21], which facilitate the development of type 2 inflammatory responses characterized by eosinophilia, airway remodeling, mucus hypersecretion, and airway hyperresponsiveness (AHR) [18–21]. The function of ILC2s must be tightly regulated in order to prevent the onset of overzealous type 2 inflammation. In this regard, several elegant studies have recently shown that interferon responses activated by viruses or microbial ligands can inhibit the ILC2-mediated type 2 inflammation [22–25]. Notably, asthmatic patients are often found to have an impaired ability to produce interferons in respopnse to common respiratory viral infections [26–28]. Genetic evidence also highlights the involvement of the interferon pathway in restraining type 2 immunopathology, e.g. single-nucleotide polymorphisms in introns or exons of IFNγ, IFNγR, STAT1 and IRF1 are associated with asthma [29–31]. Thus, it appears that interferon responses might play a negative role in regulating type 2 inflammation via acting on ILC2 cells.

Although the innate RNA-sensing pathways are known to activate innate immune responses that result in the production of large amounts of proinflammatory mediators such as IL1β, TNFα and type I intererons, their roles in the regulation of ILC2-driven acute type 2 lung inflammation remain largely unexplored. Here, we report that immune detection of immuostimulatory RNA intermediates (exRNAs) generated from cells infected by common respiratory RNA- or DNA-viruses can activate protective immune responses against IL-33- and a fungal allergen-induced lung inflammation and subsequent airway hyperresponsiveness (AHR). Mechanistically, exRNAs can activate the TLR3-Trif- and MAVS-mediated signaling pathways to suppress the proliferation and function of ILC2.

## Materials and methods

### Mice

TLR3[-/-] [32], Trif [Lps2] [33], MAVS[-/-] [34, 35] mice have been described previously. Wild type C57BL/6J, IFNAR1[-/-] and Rag1[-/-] mice were purchased from the Jackson Laboratory. The double knockouts TLR3[−/−]MAVS[−/−] mice were generated by intercrossing on campus. Briefly, mice were bred and maintained under specific pathogen-free conditions in the animal facility. Mice were housed in the standard conditions and provided with low fat diet, water and a 12-hour shift light/dark cycle. To minimize animal suffering and distress, mice were

anesthetized by isoflurane inhalation during intratracheal injections. The mice were monitored daily during intranasal injections and one day after the last injections and their health status was assessed by their behavior, appearance, hydration status, respiration, and presence/absence of any obvious pain. Their health and well-being were monitored daily by facility staff based on their appearance and behavior. At the time of harvest, mice were euthanized by carbon dioxide. All animal works were performed under the strict accordance with the recommendations in the Guide for the Care and Use of Laboratory Animals of the National Institutes of Health and the experimental protocols were approved by the Institutional Animal Care and Use Committee of the University of Texas Health San Antonio.

**Cells, viruses and reagents.** Human cell line HeLa were cultured in D-MEM supplemented with 10% FCS plus antibiotics. BEAS-2B cell (ATCC® CRL-9609™) was cultured in RPMI-1640 medium supplemented with 10% FBS plus antibiotics. All cultured cells were routinely monitored for the mycoplasma contamination. Poly(I:C) LMW (Cat. tlrl-picw) was purchased from InvivoGen. Recombinant murine IL-33 (Cat. 210–33) was purchased from Peprotech. Human rhinovirus 1B (VR-1645) and human adenovirus 5 (VR-5) were obtained from ATCC. Vaccinia virus (VACV, WR strain) and VSV (Indiana strain) have been described previously [34, 35].

**Isolation of total RNAs from virus-infected HeLa cells.** HeLa cells were infected with various viruses at 1.0 MOI at 80–90% confluence. Depending on the individual virus, it usually took 1–3 days when about half of infected cells were dead. At the time, the total RNA was purified from the attached cells by adding proper amount of TRIzol (Invitrogen).

**Detection of dsRNA with dot blot.** Spotted RNAs of HeLa cells infected by various viruses as indicated were crosslinked to Amersham Hybond-N+ membrane (GE Health) in a Stratalinker 2400 UV Crosslinker. The membrane was washed with washing buffer (TBST) and blocked in blocking buffer (5% nonfat milk in TBST) with gentle shaking. Incubate the membrane with J2 anti-dsRNA monoclonal antibody (SCICONS) overnight at 4˚C with gentle shaking. For detection, AP-conjugated anti-mouse was added at a dilution of 1:5,000.

**Transfection, allergen and nucleic acid treatment.** Transfection of RNA (1.0 μg/ml) as indicated into cultured BEAS-2B cell was carried out using Lipofectamine 3000 (Invitrogen). For enzyme treatments of nucleic acids, allergen extracts or RNA was treated with various nucleases (Invirtogen) as indicated at 37˚C for 0.5h.

**Real-time quantitative PCR.** Reverse transcription and real-time PCR (qPCR) reactions were carried out using iScript cDNA synthesis kit and iQ SYBR Green Supermix (Bio-Rad). qPCR was performed on a Bio-Rad CFX384 Touch™ Real-Time PCR Detection System using the following primers: Mouse primers (Forward:5'→3', Reverse: 5'→3'): Rpl19 (`AAATCGCCA ATGCCAACTC, TCTTCCCTATGCCCATATGC`); IL1β (`TCTATACCTGTCCTGTGTAATG, GCT TGTGCTCTGCTTGTG`); IFIT3 (`TGGCCTACATAAAGCACCTAGATGG, CGCAAACTTTTGGCA AACTTGTCT`); ISG15 (`GAGCTAGAGCCTGCAGCAAT, TTCTGGGCAATCTGCTTCTT`); Mx1 (`TCTGAGGAGAGCCAGACGAT, ACTCTGGTCCCCAATGACAG`); OASL2 (`GGATGCCTGGGAG AGAATCG, TCG CCTGCTCTTCGAAACTG`); TNFα (`CCTCCCTCTCATCAGTTCTATGG, GGCT ACAGGCTTGTCACTCG`); ZBP1 (`GCTATGACGGACAGACGTGG, TGTTGACCG GATTGTGCT GA`);. Human Primers: GAPDH (`ATGACATCAAGAAGGTGGTG, CATACCAGGAAATGAGCT TG`); IFNβ (`AGGACAGGATGAACTTTGAC, TGATAGACATTAGCCAGGAG`); IFNλ1 (`CGCCTTG GAAGAGTCACTCA, GAAGCCTCAGGTCCCAATTC`).

***In vivo* administration and FACS analysis of BALF and lung.** Mice were anesthetized by isofluorane inhalation, followed by the intra-tracheal administration with either Poly(I:C) 5 μg) or exRNAs (5 μg), rIL-33 (0.25 μg), or *A. flavus* protease allergen (10 μg) in 80 μl of PBS. Mice were sacrificed at indicated times and the trachea was catheterized and flushed with 1 ml of ice-cold PBS-EDTA three times. Differential cells in BALF were labeled with antibodies as

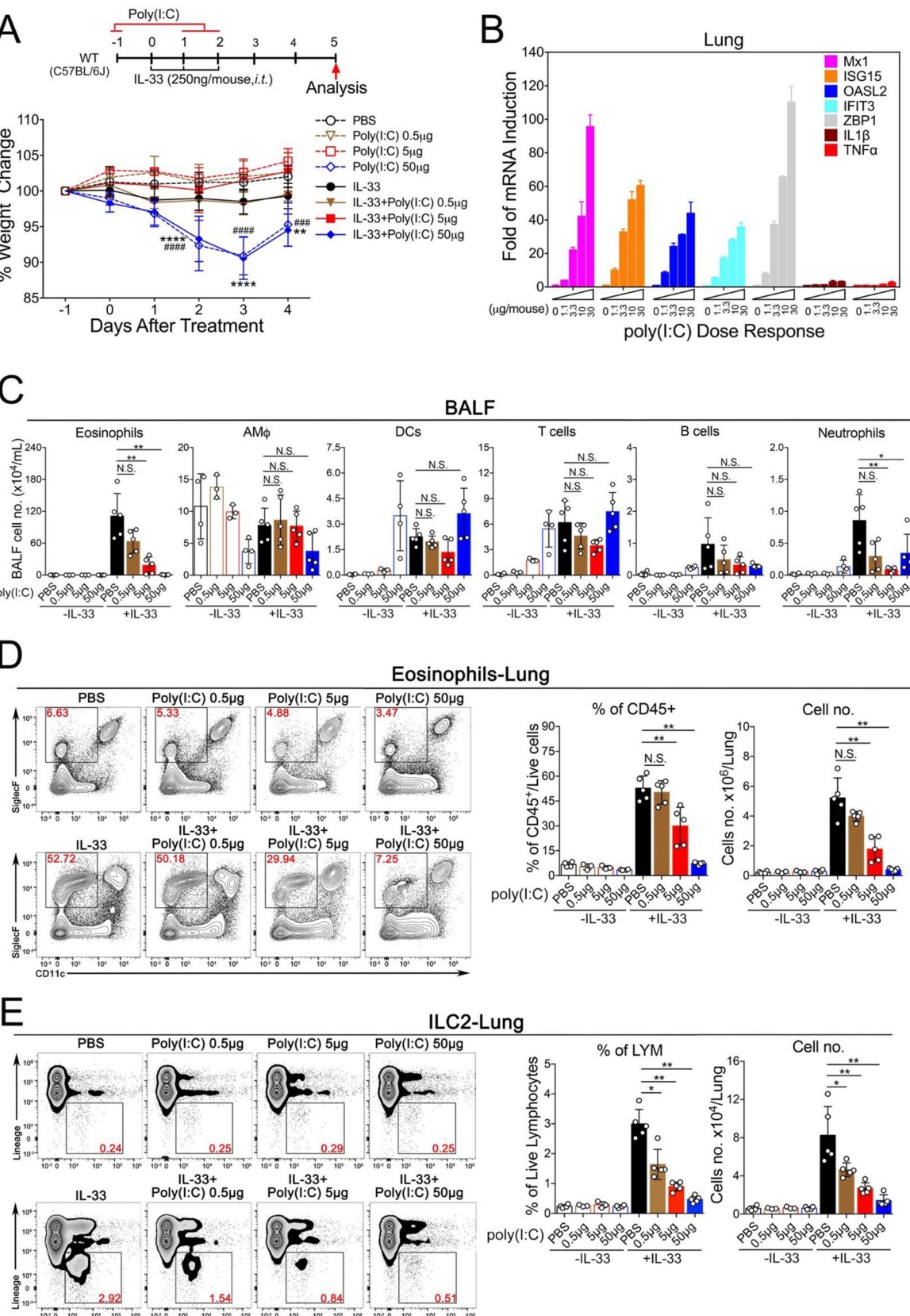

**Fig 1. Poly(I:C) activates innate immune responses in a dose-dependent manner and inhibits IL-33 induced eosinophilia and ILC2 proliferation. A.** Experimental setup of an acute lung inflammation model that illustrates the animal groups, the corresponding treatment regimen and timeline (upper panel) and weigh changes following treatment as indicated. Error bars represent SEMs. Statistics of the weight changes was determined using the Two-Way ANOVA. The dot lines (animal groups treated with either PBS or increased amount of Poly(I:C) without IL-33, [###] p < 0.001, [####] p < 0.0001); the sold lines (animal groups treated with PBS or increased amount of Poly(I:C) with IL-33, [**] p < 0.01, [****]p < 0.0001). **B.** Transcriptional induction of gene expressions by Poly(I:C) in mouse lungs. **C.** Groups of mice as indicated were treated with PBS, IL-33 in presence of increased doses of Poly(I:C) as indicated. Bronchoalveolar lavage fluid (BALF) was collected and analyzed for differential immune cell types. The result is a pool of two independent experiments. (n = 3–5 per group as indicated with open circles). **D.** Administration of Poly(I:C) in a dose-dependent manner reduced the percentage and number of lung-eosinophils after exposure to IL-33. (n = 3–5 per group as indicated with open circles). **E.** Poly(I:C) treatment decreased the number of lung-ILC2 cells after exposure to IL-33 in a dose-dependent manner. (n = 3–5 per group as indicated with open circles). Unless otherwise indicated, P value was determined using Mann-Whitney test, P value ≥0.05 was not considered statistically significant [N.S.]. [*] p < 0.05, [**] p < 0.01.

indicated, then mixed with counting beads (Spherotech) for further FACS analysis on a BD Celesta cell analyzer. Flow cytometry data were analyzed using FlowJo software. The antibodies and reagents for FACS analysis are: SPHERO™ AccuCount Fluorescent Particles (Spherotech, Cat. #ACFP-70-5); Anti-Mouse Siglec-F PE (clone E50-2440) (BD Bioscience, Cat. #552126); Anti-Mouse CD19 Alexa Fluor® 647 (clone 1D3) (BD Bioscience, Cat. #557684); Anti-Mouse CD3ε APC (clone 145-2C11) (BioLegend, Cat. #100322); Anti-Mouse MHC II APC-Cy7 (clone M5/114.15.2) (BioLegend, Cat. #10627); Anti-Mouse CD11c PE-Cy7 (clone N418) (TONBO bioscience, Cat. #60-0114-U100); Anti-Mouse CD11b V450 Rat (clone M1/70) (BD Bioscience, Cat. #560456); Anti-Mouse Ly-6G FITC (clone RB6-8C5) (Invitrogen, Cat. #11-5931-82); Anti-Mouse Fixable Viability Dye eFluor 506 (Invitrogen, Cat. #65-0866-14); Anti-Mouse CD45 APC-Cy7 (clone 30-F11) (BD Bioscience, Cat. #561037).

**Identification of lung ILC2.** Lung ILC2 identification was performed as described previously [36]. Lung tissues were digested in 8 ml RPMI-1640 containing Liberase (50 μg/ml) and DNase I (1 μg/ml) for about 40 min at 37˚C. Cell suspensions were filtered through 70 μm cell strainers and washed once with RPMI-1640. For ILC2 identification, total lung cell suspensions were blocked with 2.4G2 antibodies and stained with lineage cocktail mAbs (CD3ε, CD4, CD8α, CD11c, FceRIα, NK1.1, CD19, TER119, CD5, F4/80, Gr-1, and Ly6G), PE-conjugated T1/ST2, PerCP-Cy5.5-conjugated CD25, V450-conjugated Sca-1, PE-Cy7-conjugated KLRG1, APC-Cy7-conjugated CD45 and eFluor 506 Fixable Viability Dye.

**Intranuclear and intracellular staining.** Intranuclear staining for Ki-67 and transcription factors was performed with the True-Nuclear™ Transcription Factor Buffer Set kit (BioLegend) according to the manufacturer's protocol. For intracellular cytokine staining, single-cell suspensions from the lungs of mice were prepared with Liberase™ and DNase I. $2 \times 10^6$ total live nucleated cells were stimulated in 200 μl RPMI-1640 media containing 10% FBS, Penicillin/Streptomycin (P+S), 2-mercaptoethanol (2-ME), brefeldin A (GolgiPlug, BD Biosciences) and PMA (phorbol 12-myeistate 13-acetate) (30 ng/ml) at 37˚C for 3 hours. After surface staining, cells were fixed and permeabilized with BioLegend Cytofix/Perm buffer and further stained intracellularly with anti-mouse IL-5 and IL-13. Dead cells were stained with eFluor506 Fixable Viability Dye before fixation and permeabilization and excluded during analysis.

**Lung inflammation and pathology.** Each mouse was intra-tracheally administered IL-33 or *A. flavus* on Day 0, 1 and 2 with or without Poly(I:C) or exRNAs on Day -1, 1 and 2 as shown in the relevant experimental diagram. There was an 8 hours interval between two treatments within the same day. Two days after the last challenge (Day 5), lung tissues were taken and fixed in 4% paraformaldehyde, paraffin embedded, cut into 4-μm sections, and stained with hematoxylin and eosin (H&E) and periodic acid-Schiff (PAS). Complete images of control and treated lungs were obtained digitally using the Aperio Scanscope XT (Aperio, Vista,

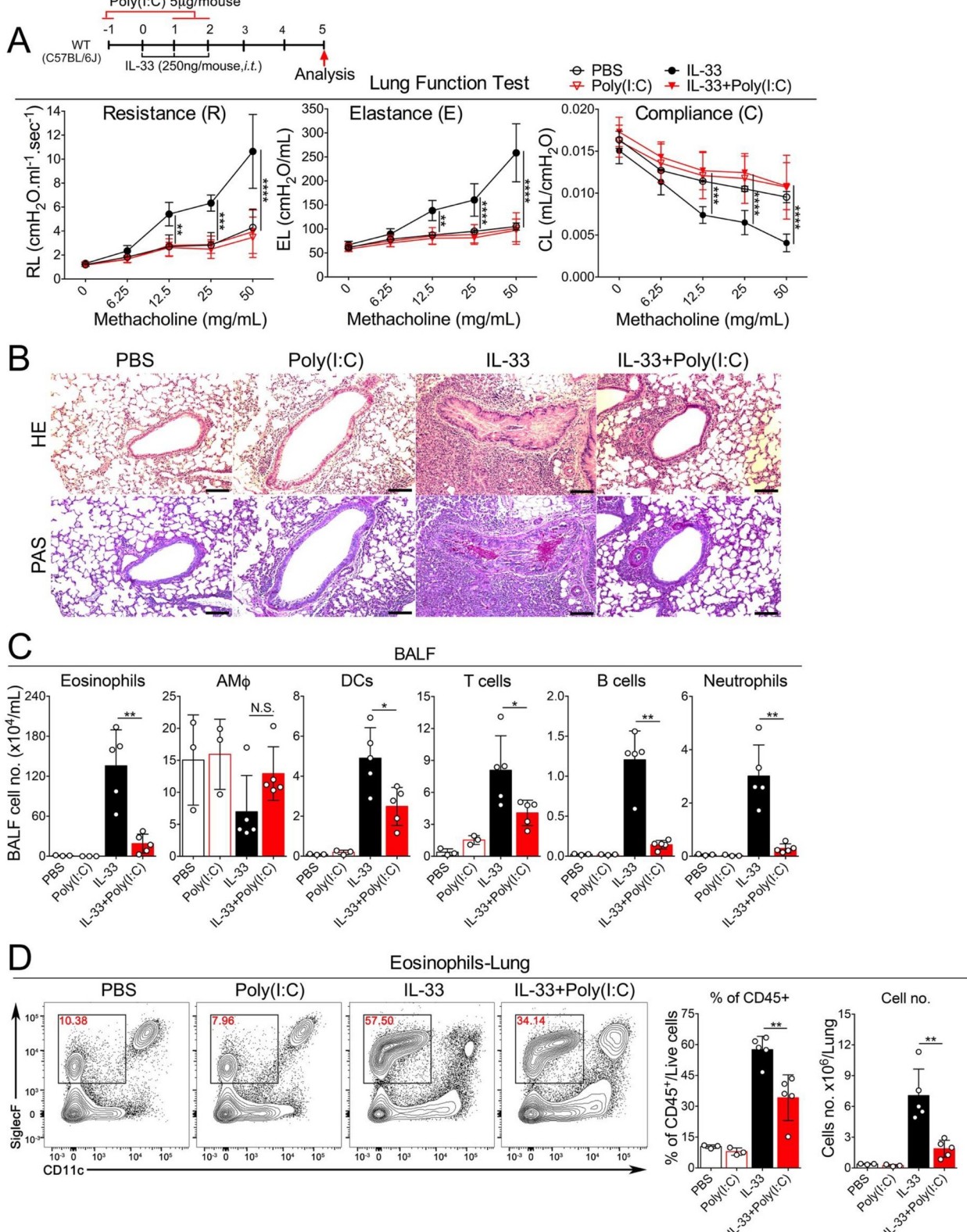

**Fig 2. Poly(I:C) at a low dose inhibits IL-33-induced AHR and acute eosinophilia. A.** Groups of mice as indicated were treated with PBS, IL-33 with or without Poly(I:C) (5 μg/mouse) as indicated. Lung function was examined by Flexivent (Scireq). Airway resistance (**R**), elastance (**E**) and compliance (**C**) were measured after exposure to increasing doses (6.25–50 mg/mL) of aerosolized methacholine. (n = 3–5, P value <0.05 was

considered statistically significant, 2way ANOVA, Tukey's multiple comparisons test, $^*$ p < 0.05, $^{**}$ p < 0.01, $^{***}$ p < 0.001, $^{****}$ p < 0.0001). **B.** Lung pathology was assessed with H&E and PAS staining. Representative images are shown here (scale bars, 100 μm). **C.** Bronchoalveolar lavage fluid (BALF) was collected and analyzed for differential immune cell types. The result is a pool of two independent experiments. (n = 3–5 per group as indicated with open circles). **D.** Administration of Poly(I:C) decreased the percentage and number of eosinophils in lungs after exposure to IL-33. (n = 3–5 per group as indicated with open circles).

CA). Printed images of lungs from both study groups were graded for disease severity using a panel of standards as previously described [37].

**Measurement of pulmonary function.** In the IL-33- and *A. flavus*-induced lung inflammation model, changes in mouse pulmonary function after allergen exposure were determined by invasive measurements using the Flexivent system (Scireq, Montreal, PQ, Canada). On day 5, the trachea was intubated after anesthetization. The lungs were mechanically ventilated. Indicators of airway hyperreactivity (AHR), including airway resistance (R), elastance (E) and compliance (C), were measured after increasing doses (6.25–50 mg/ml) of aerosolized methacholine.

**Statistical analysis.** The statistical analysis was done using software GraphPad Prism 6. For comparison of two groups, unless otherwise indicated, P values were determined by Mann-Whitney test. For comparison of more than two groups, Two-Way ANOVA was performed. P value<0.05 was considered statistically significant. P values are indicated on plots and in figure legends. (P value ≥0.05 was not considered statistically significant [N.S.], $^*$ p < 0.05, $^{**}$ p < 0.01, $^{***}$ p < 0.001, $^{****}$ p < 0.0001).

**Raw data availability.** All relevant data are included in the main figure and the associated Supporting Information files. All raw data files are available to readers upon request.

## Results

### Poly(I:C) inhibits acute type 2 lung inflammation in a dose-dependent manner

Although it has been previously demonstrated that the synthetic dsRNA analogue Poly(I:C) can exacerbate antigen-induced respiratory allergic responses [38–40], its effect in acute models of type 2 lung inflammation remains unknown. To address this issue, we used the recombinant murine protein IL-33 to elicit an acute lung inflammation [18, 41] (**Fig 1A, upper panel**). To investigate whether the respiratory delivery of Poly(I:C) at various doses, alone or in combination with IL-33, could cause systemic immune toxicity, we measured the bodyweight of the treated animals (**Fig 1A, lower panel**). No significant weight changes were observed in animals treated with PBS or Poly(I:C) at 0.5 or 5 μg/mouse alone or in combination with IL-33, which contrasted to the groups treated with Poly(I:C) at 50 μg/mouse. Next, to examine whether Poly(I:C) could induce an immune response in the lungs, mice were administered with increased doses of Poly(I:C) followed by examination of gene expressions by Real-Time quantitative PCR (RT-qPCR). The expression of multiple ISGs (Mx1, ISG15, OASL2, IFIT3 and ZBP1) were upregulated (**Fig 1B**). Functionally, Poly(I:C) treatment in a dose-dependent manner led to the dramatic reduction of eosinophils and ILC2 cells in bronchoalveolar lavage fluid (BALF) and lungs (**Fig 1C–1E,** the FACS gating strategy is shown in **S1 Fig**). Since Poly(I:C) at 5 μg/mouse did not cause either systemic toxicity or any apparent up-regulation of two proinflammatory genes IL1β and TNFα, we chose this dose for the subsequent *in vivo* experiments in combination with IL-33. Taken together, these results suggest that poly(I:C) treatment at a relatively low dose (5 μg/mouse) triggers a protective immune response in mouse lungs that is sufficient to prevent the development of IL-33-driven lung eosinophilia.

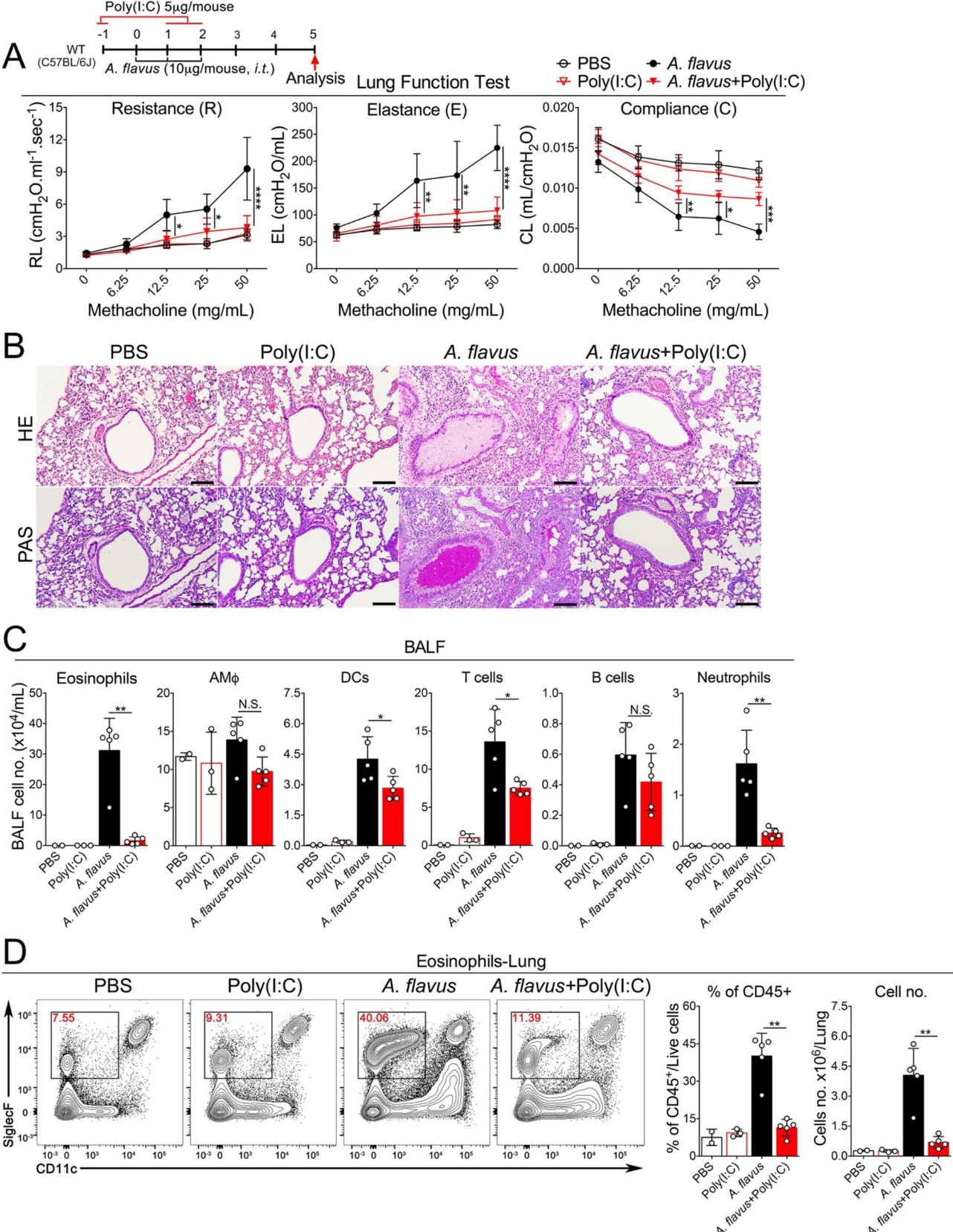

**Fig 3. Poly(I:C) at a low dose inhibits *A. flavus* induced AHR and acute eosinophilia. A.** Groups of mice as indicated were treated with PBS, Poly(I:C), *A. flavus* or *A. flavus*+Poly(I:C). Airway hyperreactivity (AHR) was examined by Flexivent (Scireq). Airway resistance (**R**), elastance (**E**) and compliance (**C**) were measured after exposure to increasing doses (6.25–50 mg/mL) of aerosolized methacholine. (n = 2–5 per group as indicated with open circles, P value <0.05 was considered statistically significant, 2way ANOVA, Tukey's multiple comparisons test, * p < 0.05, ** p < 0.01, *** p < 0.001, **** p < 0.0001). **B.** Lung pathology was assessed with H&E and PAS staining. Representative images are shown here (scale bars, 100 μm). **C.** BALF was collected and analyzed for differential immune cell types. The result is a pool of two independent experiments. (n = 2–5 per group as indicated with open circles). **D**. Administration of Poly(I:C) decreased the percentage and number of eosinophils in lungs after exposure to *A. flavus*. (n = 2–5 per group as indicated with open circles).

## Poly(I:C) at a low dose ameliorates both IL-33- and *A. flavus*-induced induced AHR and lung immunopathology

After determining the optimal effective dose of Poly(I:C), we went on to perform more detailed analysis using two models of acute type 2 lung inflammation. In addition to IL-33, we also included a clinically relevant model of allergic inflammation caused by a fungal allergen, *A. flavus*, that has been shown to act through the IL-33-mediated pathway [42]. After mice were challenged with either IL-33 or the extract of *A. flavus*, we found that Poly(I:C) treatment at 5 μg/mouse dramatically improved the lung function evidenced by the reduced AHR, elastance, and improved compliance (**Figs 2A & 3A**). Consistently, Poly(I:C) treatment ameliorated IL-33- and *A. flavus*-induced lung pathologies characterized by infiltration of inflammatory cells, mucus overproduction and epithelial cell hyperplasia (**Figs 2B & 3B**). We also found that markers of IL-33 or *A. flavus*–driven type 2 inflammation were greatly reduced, as evidenced by the significant decrease in eosinophils in the BALF and lungs of Poly(I:C)-treated mice (**Figs 2C, 2D & 3C, 3D**). Thus, these data indicate that Poly(I:C) treatment can activate a protective response to attenuate acute type 2 lung inflammation induced by both the type 2 inducer cytokine IL-33 and a clinically relevant fungal allergen *A. flavus*.

## Poly(I:C) at a low dose inhibits the activation and proliferation of lung ILC2

Next, we determined whether Poly(I:C) could affect the function and proliferation of lung ILC2 cells *in vivo* when exposed to IL-33 or *A. flavus*. To rule out the involvement of adaptive immunity, in addition to wild type (WT) C57BL/6J mice, Rag1$^{-/-}$ mice were tested with IL-33+Poly(I:C) (**Fig 4A**). The gating strategy for calculating the number of lung-ILC2s is shown in **S2 Fig**. ILC2 expansion was significantly suppressed, as determined by the reduction in the percentage and number of total lung-ILC2 in both strains of mice. Functionally, the percentage of IL-33 activated ILC2s (IL-5$^+$/IL-13$^+$ double-positive) was significantly decreased upon treatment of Poly(I:C) (**Fig 4B and 4D**) in both WT and Rag1$^{-/-}$ mice. Consistent with previous experiments, eosinophils in the BALF and lungs were significantly reduced in Poly(I:C)-treated wild type and Rag1$^{-/-}$ mice (**S3 Fig**) compared to their counterparts when exposed to IL-33. Further, Poly(I:C)-mediated inhibitory effects on ILC2 proliferation in wild type mice was confirmed by Ki-67 staining (**Fig 4E**). We continued to examine whether Poly(I:C) treatment regulates ILC2 function in the context of *A. flavus* exposure. FACS analysis revealed that while *A. flavus* significantly increased the percentages of IL5$^+$/IL13$^+$ double-positive cells and the total numbers of lung ILC2 cells, co-administration of Poly(I:C) abolished this induction (**Fig 4C**). Taken together, these results strongly suggest that Poly(I:C) activates an innate immune signaling pathway to negatively regulate ILC2-induced eosinophilic lung inflammation and that this suppression was independent of adaptive immunity in mice.

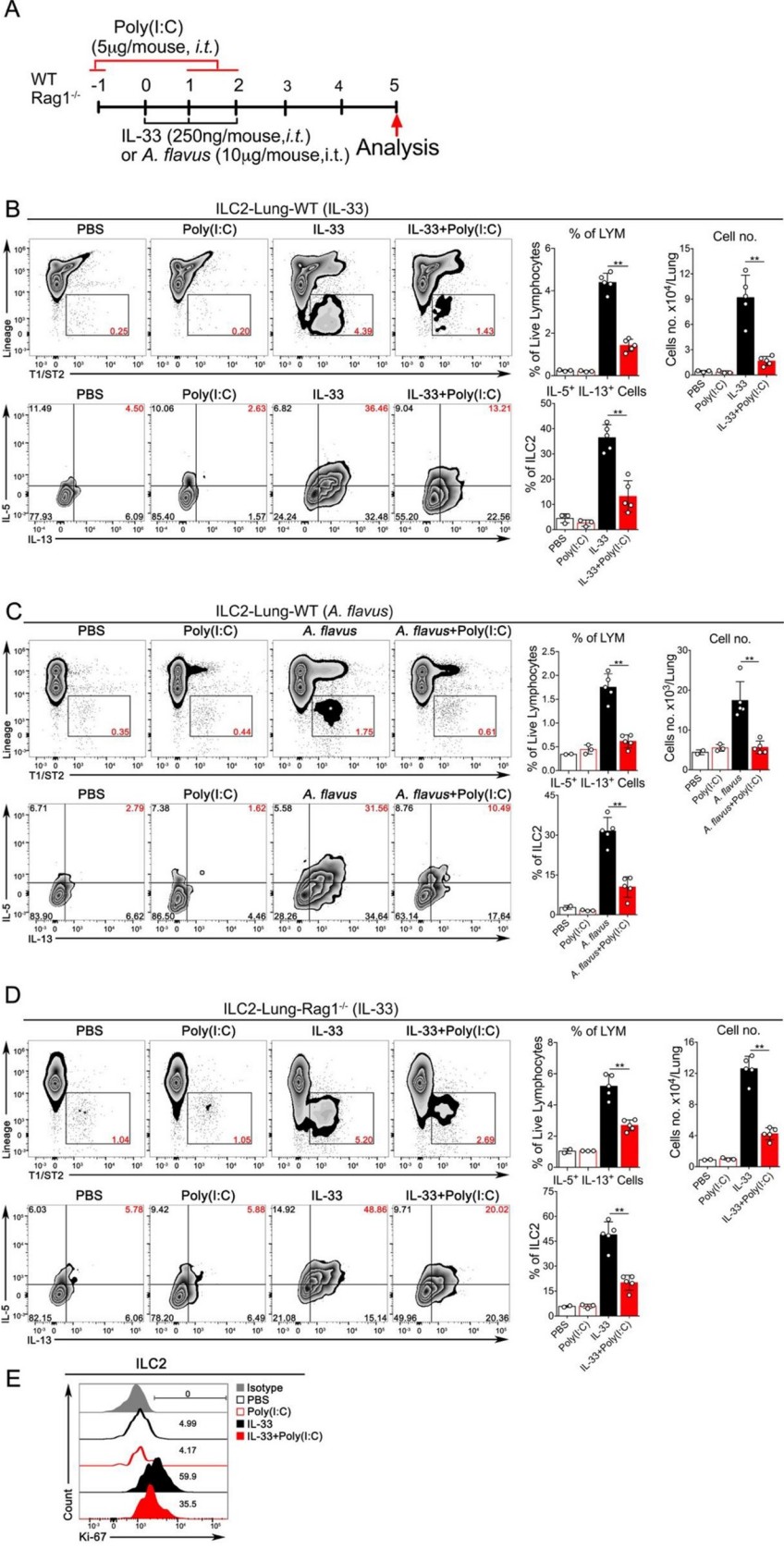

**Fig 4. Poly(I:C) inhibits the activation and proliferation of ILC2 cells induced by IL-33 or *A*. *flavus* in WT and Rag1[-/-] mice. A.** Experimental protocol showing the animal groups (WT and Rag1[-/-]), the corresponding treatment regimen and timeline. **B.** C57B6/J WT mice were treated with PBS, Poly(I:C), IL-33 or IL-33+Poly(I:C). Lung single cell suspensions were prepared and the percentage and number of ILC2 cells in lungs were analyzed. In addition, lung cells were stimulated with PMA in cultures as described in the Materials and Methods. The percentage of IL5[+]/IL13[+]-double positive ILC2 cells in lungs were analyzed. (n = 3–5 per group as indicated with open circles). **C.** Similar to **B**, instead of IL-33, *A*. *flavus* was used. (n = 2–5). **D.** Similar to **B**, instead of WT mice, the Rag1[-/-] mice were used. (n = 2–5 per group as indicated with open circles). **E.** C57B6/J WT mice were treated with PBS, Poly(I:C), IL-33 or IL-33+Poly(I:C). The lung ILC2 cells were analyzed with Ki-67 staining and isotype antibody. The result is a representative of three independent experiments.

## The TLR3-Trif-IFNAR1 signaling axis mediates the inhibitory effects of Poly(I:C) on IL-33-induced acute lung inflammation

Activation of RNA-sensing pathways usually results in the production of multiple effector molecules such as type I interferons (IFN-I) and other proinflammatory cytokines such as TNFα and IL1β, which may function together to modulate ILC2-driven type 2 inflammatory response. To determine the role of the TLR3-Trif-IFNAR1 signaling axis, we performed a series of experiments using mice genetically deficient in TLR3, Trif or IFNAR1. Remarkably, we found that in the context of IL-33-induced acute lung inflammation, the inhibitory effects of Poly(I:C) were completely abolished in these three knockout strains. More importantly, in contrast to wild type mice shown in **Fig 2**, TLR3[-/-] (**Fig 5A & 5B**), Trif [Lps2] (**Fig 6A & 6B**) and IFNAR1[-/-] mice (**Fig 7A & 7B**) treated with either IL-33 or IL-33+Poly(I:C) did not show any significant changes in eosinophils (number and percentage) in BALF or lungs, and the activation of lung-ILC2 cells were unaffected either (**Figs 5C, 6C and 7C**). Collectively, these results suggest that TLR3-Trif-IFNAR1 signaling axis mediates the effector function of Poly(I:C) to negatively regulate IL-33-induced type 2 lung inflammation via suppressing the activation of lung ILC2 cells.

## Extrinsic sensing of natural RNAs of virus-infected cells prevents *A*. *flavus*-induced type 2 lung immunopathology

Poly(I:C) is widely used to imitate viral infections. However, Poly(I:C) is a chemically synthesized RNA analog, not a natural RNA. Although we showed above that Poly(I:C) effectively prevented ILC2-mediated immunopathology induced by IL-33 and *A*. *flavus*, it remains unclear whether natural RNAs from virus-infected cells could function similarly to Poly(I:C) *in vivo* to inhibit the outcome of allergen-induced acute type 2 inflammation. To address this issue, we isolated total RNAs from either RNA- or DNA-virus infected cells and tested their immunostimulatory activities *in vitro* and *in vivo*. We found that long dsRNA species (> 40bp) were detected in total RNAs isolated from both RNA- and DNA-virus infected cells including human rhinovirus (RV1B), vesicular stomatitis virus (VSV), human adenovirus (AdV5) and vaccinia virus (VACV) using a Dot-blot with dsRNA-specific J2 monoclonal antibody, which is consistent with previous reports [43–46] (**S4A Fig**). Functionally, these natural vRNAs were able to induce the expression of type I & III interferons in a lung epithelial cell BEAS-2B (**S4B Fig**). When delivered into mouse lungs, both RV1B- and AdV5-RNAs potently stimulated the expression of ISGs in a dose-dependent manner (**S4C Fig**). More importantly, at the same dose of Poly(I:C), the treatment with either RV1B RNA or AdV5 RNA was sufficient for reducing the infiltration of airway and lung eosinophils and suppressing the activation of lung ILC2 cells in wild type mice while the inhibitory effects of RV1B RNA were completely abolished in TLR3[-/-]MAVS[-/-] mice (**Figs 8A–8C, 9A–9C**), but not in TLR3[-/-] mice (**S5 Fig**), indicating that compared to Poly(I:C) that solely depended on the TLR3-Trif

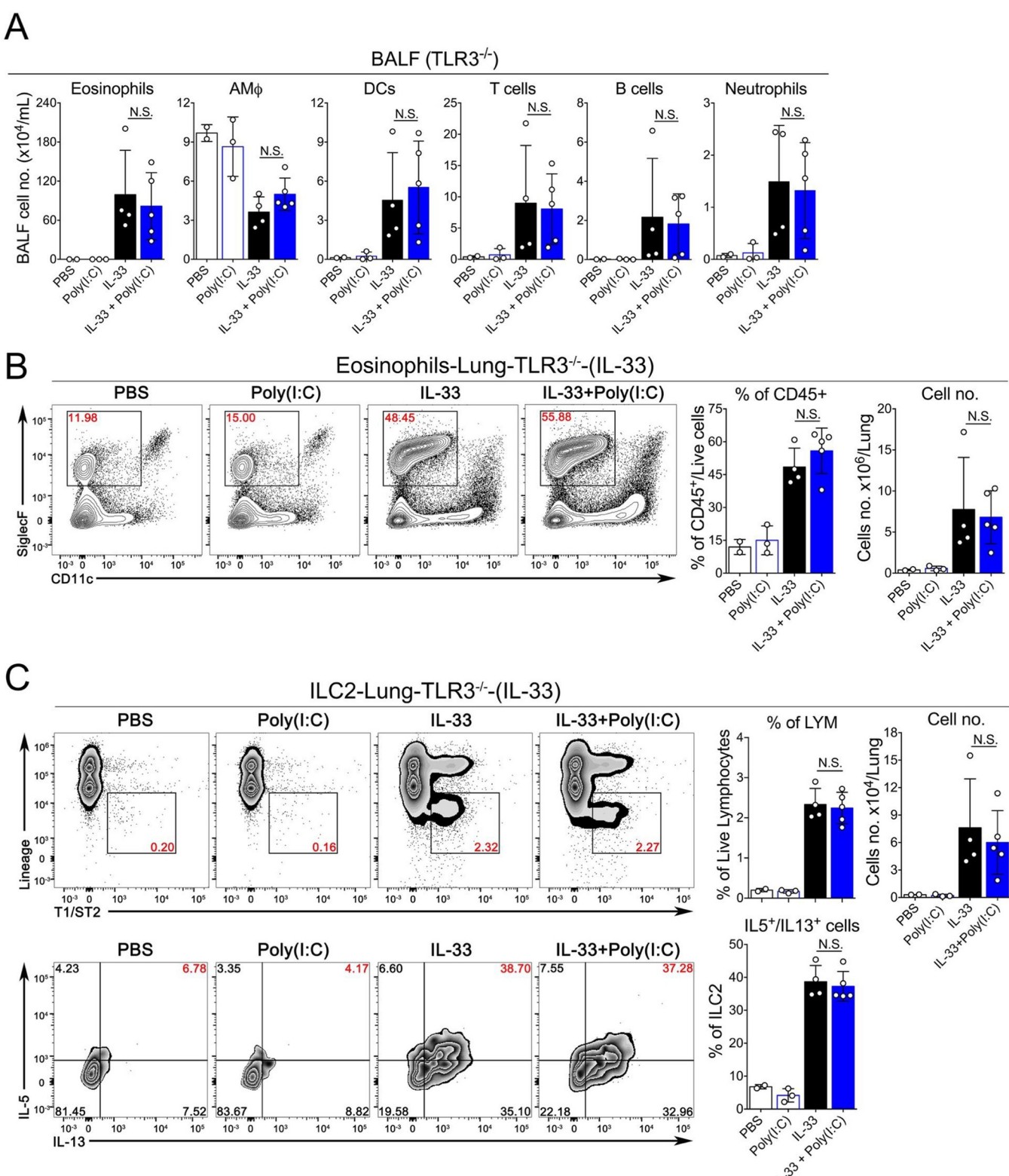

**Fig 5. The inhibitory effect of Poly(I:C) is mediated by the TLR3-mediated pathway. A.** Groups of TLR3[-/-] mice as indicated were treated with PBS, Poly(I:C), IL-33 or IL-33+Poly(I:C). BALF was collected and analyzed for differential immune cell types. (n = 2–5 per group as indicated with open circles, P value ≥0.05 was not considered statistically significant [N.S.]). **B.** Administration of Poly(I:C) into TLR3[-/-] mice did not significantly decrease the percentage and number of eosinophils in lungs after exposure to IL-33. (n = 2–5 per group as indicated with open circles, P value ≥0.05 was not considered statistically significant [N.S.]). **C.** The percentage and number of ILC2 cells and percentage of IL5[+]/IL13[+]-double positive ILC2 cells in lungs of TLR3[-/-] mice were analyzed.

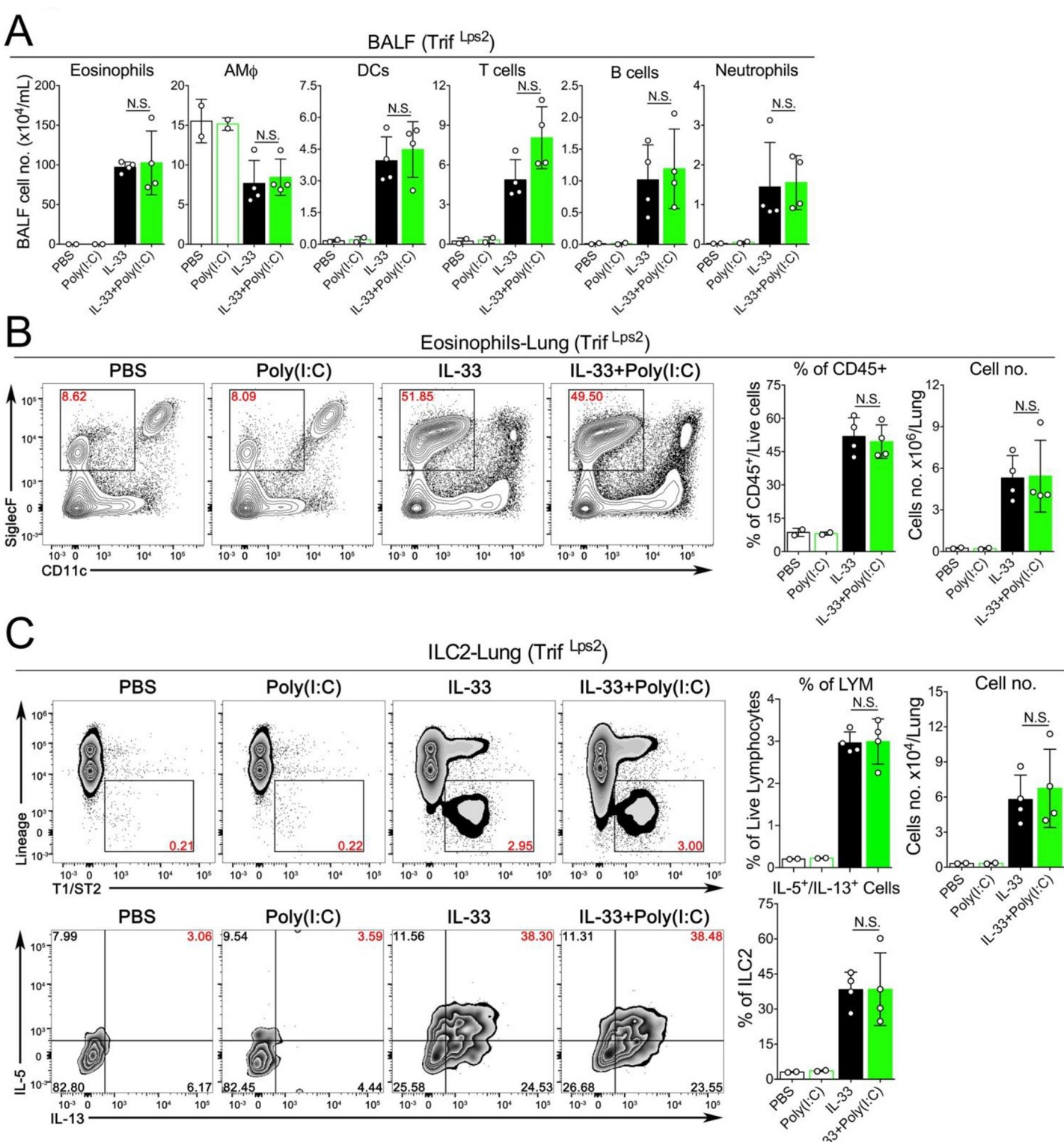

**Fig 6. The inhibitory effect of Poly(I:C) is mediated by the Trif-mediated pathway. A.** Groups of Trif[Lps2] mice as indicated were treated with PBS, Poly(I:C), IL-33 or IL-33+Poly(I:C). BALF was collected and analyzed for differential immune cell types. (n = 2–4 per group as indicated with open circles, P value ≥0.05 was not considered statistically significant [N.S.]). **B.** Administration of Poly(I:C) into Trif[Lps2] mice did not significantly decrease the percentage and number of eosinophils in lungs after exposure to IL-33. (n = 2–4 per group as indicated with open circles, P value ≥0.05 was not considered statistically significant [N.S.]). **C.** The number and percentage of IL5[+]/IL13[+]-double positive ILC2 cells in lungs of Trif[Lps2] mice were analyzed.

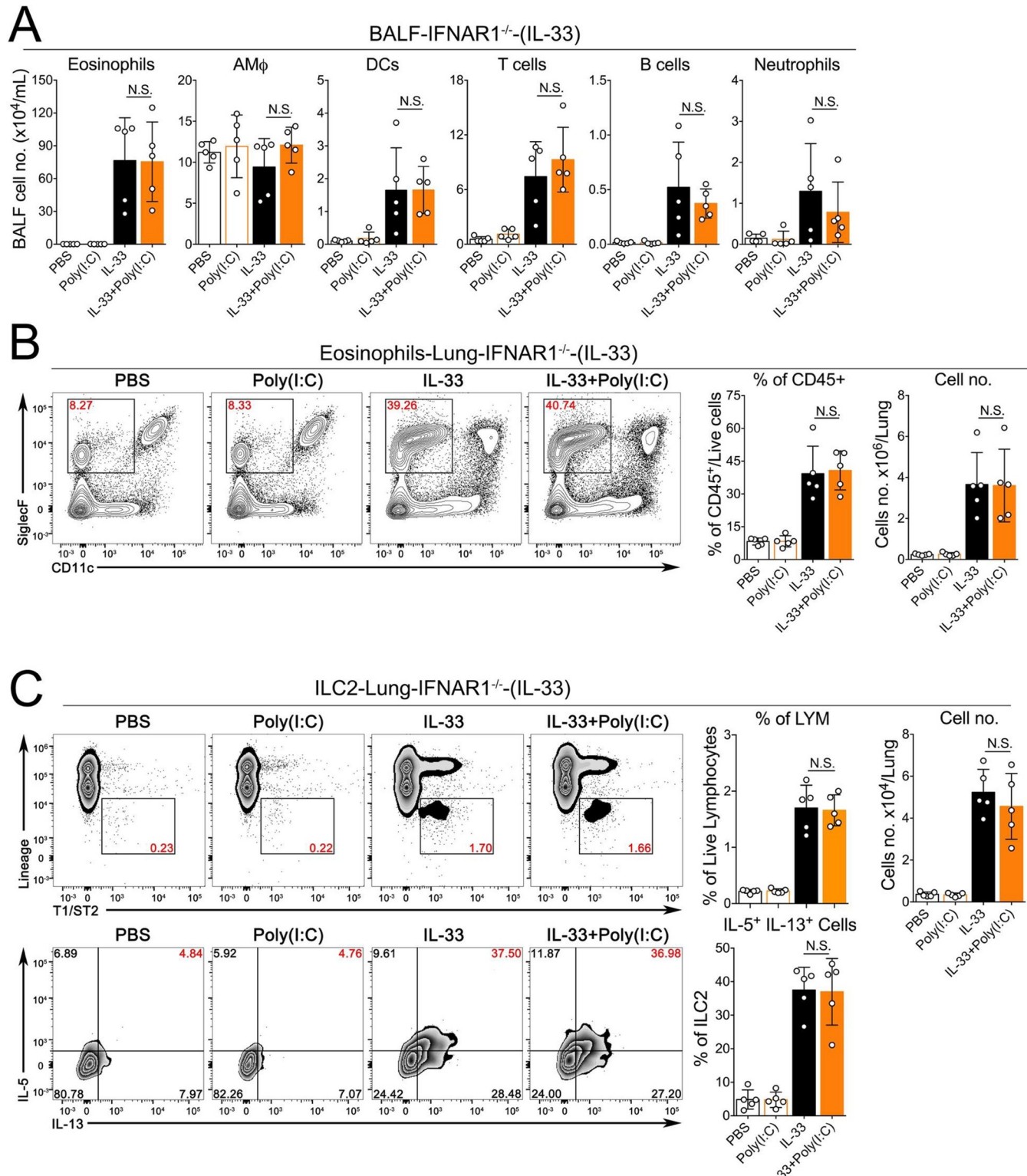

**Fig 7. The inhibitory effect of Poly(I:C) is mediated by the IFN-I signaling pathway. A.** Groups of IFNAR1[-/-] mice as indicated were treated with PBS, Poly (I:C), IL-33 or IL-33+Poly(I:C). BALF was collected and analyzed for differential immune cell types. (n = 5 per group as indicated with open circles, P value ≥0.05 was not considered statistically significant [N.S.]). **B.** Administration of Poly(I:C) into IFNAR1[-/-] mice did not significantly decrease the percentage and number of eosinophils in lungs after exposure to IL-33. (n = 5 per group as indicated with open circles, P value ≥0.05 was not considered statistically significant [N.S.]). **C.** The percentage and number of ILC2 cells and percentage of IL5[+]/IL13[+]-double positive ILC2 cells in lungs of IFNAR1[-/-] mice were analyzed.

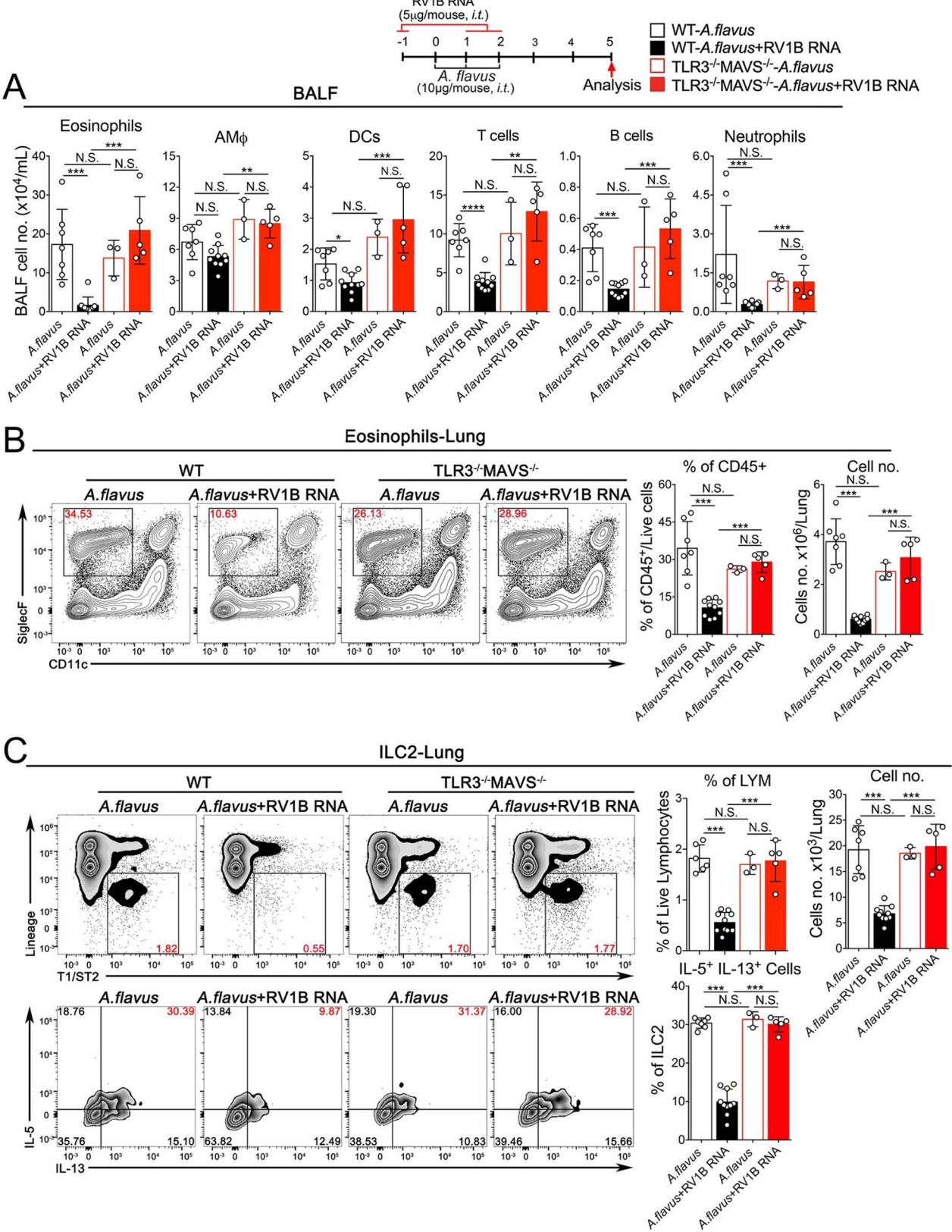

**Fig 8. exRNAs from RV1B-infected cells activates innate RNA-sensing pathways to prevent *A. flavus*-induced lung inflammation. A.** Groups of WT and TLR3$^{-/-}$MAVS$^{-/-}$ mice as indicated were treated with *A. flavus* or *A. flavus*+RV1B RNA. BALF was collected and analyzed for differential immune cell types. (n = 3–10 per group as indicated with open circles, P value $\geq$0.05 was not considered statistically significant [N.S.]). **B.** In contrast to WT mice, administration of RV1B RNA into TLR3$^{-/-}$MAVS$^{-/-}$ mice did not significantly decrease the percentage and number of eosinophils in lungs after exposure to A. *flavus*. (n = 3–10 per group as indicated with open circles, P value $\geq$0.05 was not considered statistically significant [N.S.]). **C.** The percentage and number of ILC2 cells and percentage of IL5$^+$/IL13$^+$-double positive ILC2 cells in lungs of WT and TLR3$^{-/-}$MAVS$^{-/-}$ mice were analyzed (P value $\geq$0.05 was not considered statistically significant [N.S.]).

signaling axis in mouse lungs as shown in **Figs 5 & 6,** RV1B RNA seemed to activate both TLR3- and MDA5-MAVS-mediated signaling pathways, similar to RV1B infection [47]. Collectively, these results suggest that viral RNAs derived from cells infected by RNA- and DNA viruses seem to function similarly to Poly(I:C), by activating innate immune responses to negatively regulate type 2 lung inflammation via suppressing the activation of lung ILC2 cells. Based on these findings, we propose a working model for extracellular RNAs released from cells infected by respiratory RNA- or DNA-viruses, which may stimulate lung innate immunity to modulate ILC2-mediated type 2 inflammation as depicted in **Fig 10**. Mechanistically, exRNAs activate the TLR3-Trif signaling axis to induce the local production of type I interferons, which may act on the IFNAR1 of lung ILC2 cells to restrain their abilities to proliferate and produce type 2 cytokines in the context of exposure to IL-33 and environmental allergens such as *A. flavus*.

## Discussion

In this study, we revealed a new role for exRNA-activated innate immune responses in negatively regulating ILC2-mediated acute type 2 lung immunopathology and AHR. Mechanistically, we showed that the protective effects of exRNAs were mediated by the TLR3- and MAVS-mediated signaling axis and were independent of adaptive immunity.

In susceptible individuals, the exposure to common harmless allergens elicits aberrant type 2 responses that promote the development of allergic diseases. In contrast, these exposures usually do not lead to pathological responses in most non-atopic healthy individuals, suggesting the possible existence of anti-allergic pathways that are still poorly understood. It has been recently reported that some environmental exposures to endotoxin, can protect us against the development of allergy [5, 6], indicating that humans have evolved protective pathways to respond to environmental stimuli. Our data imply that in normal individuals, innate RNA-sensing pathways might be a homeostatic mechanism to modulate mucosal immunity against developing unrestrained type 2 inflammatory responses during environmental allergen exposures.

Our traditional perception is that respiratory viral infections result in pathological responses. Emerging evidence from animal models have suggested otherwise that some viral infections, e.g. gammaherpesvirus or influenza virus could activate innate immune responses that are protective against allergen-induced type 2 immunopathology [25, 48]. Additionally, a beneficial role of enteric viruses has also been recently reported by several studies that innate RNA sensing immunity seems to play an important function in sustaining intestinal homeostasis via educating intestinal intraepithelial lymphocytes (IELs) [49–51]. Although the exact underlying molecular mechanism remains to be uncovered, it is plausible that the protective immunity could be mediated, at least in part, by exRNAs, which has been shown to be produced in cells infected by both DNA and RNA viruses [43–46].

A few recent studies have shown that type I interferons directly inhibit the function and proliferation of ILC2 *in vitro* and *in vivo* during influenza A infection or treatment with TLRs agonists such as CpG-DNA and R848 [22, 23, 25, 52–54]. Notably, unlike RV1B and AdV5

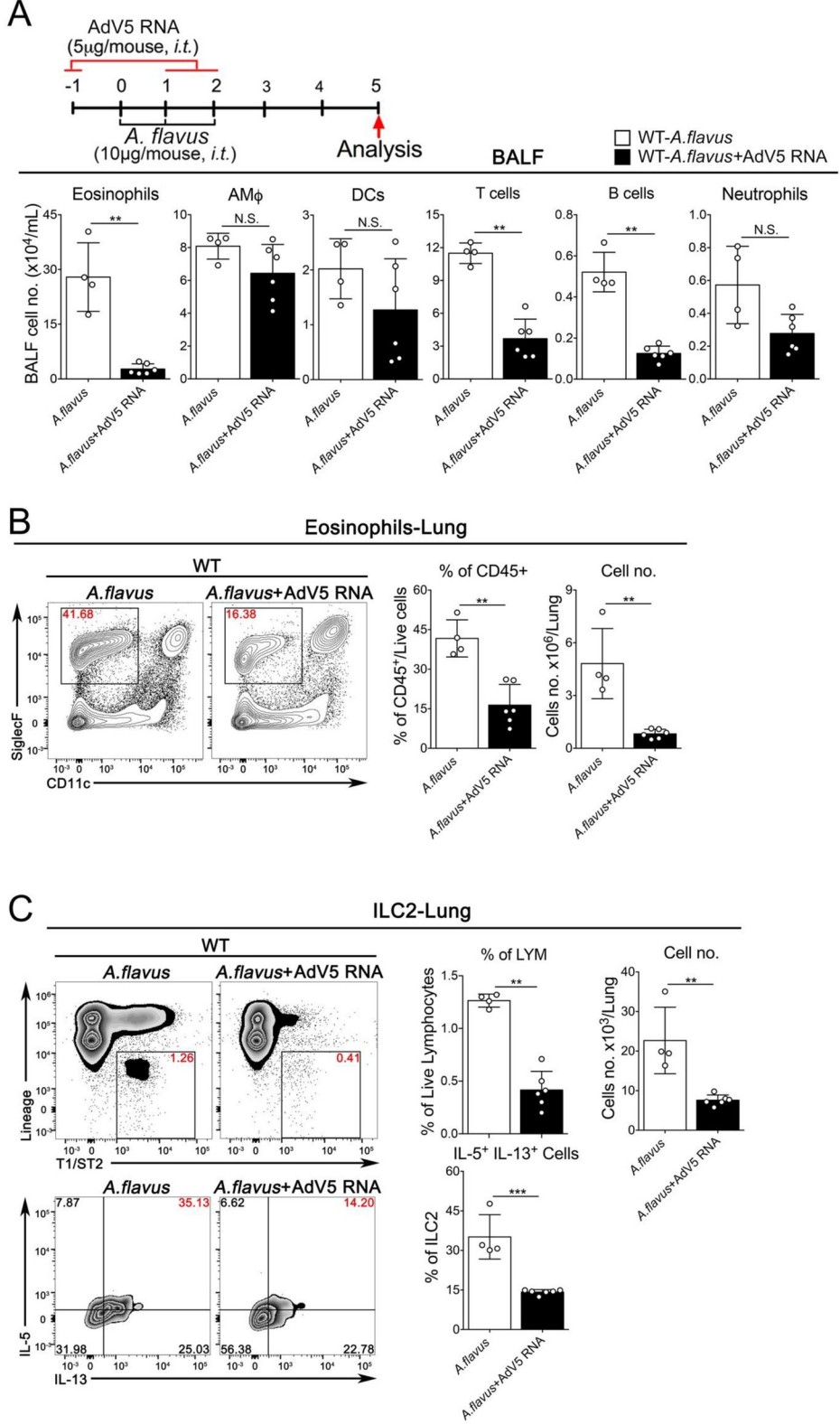

**Fig 9. exRNAs from AdV5-infected cells activates innate RNA-sensing pathways to prevent *A. flavus*-induced lung inflammation. A.** Groups of WT mice as indicated were treated with *A. flavus* or *A. flavus*+AdV5 RNA. BALF was collected and analyzed for differential immune cell types. (n = 4–6 per group as indicated with open circles, P value ≥0.05 was not considered statistically significant [N.S.]). **B.** Administration of AdV5 RNA into WT mice

significantly reduced the percentage and number of eosinophils in lungs after exposure to A. *flavus*. (n = 4–6 per group as indicated with open circles, P value ≥0.05 was not considered statistically significant [N.S.]). **C.** The number and percentage of IL5+/IL13+-double positive ILC2 cells in lungs of WT mice were analyzed (n = 4–6 per group as indicated with open circles, P value ≥0.05 was not considered statistically significant [N.S.]).

[43–46], influenza A virus did not produce detectable amounts of dsRNA in cells [43]. Thus, influenza A virus is mainly recognized by RIG-I or TLR7, but not by the long dsRNA sensors TLR3 or MDA5, to activate innate immune responses [43, 55]. Extracellular dsRNAs could be generated during common respiratory infections by both RNA and DNA viruses such as human rhinovirus and adenovirus, which usually cause mild self-limiting respiratory infections in most individuals [56, 57]. As shown in **S4C Fig**, extracellular viral dsRNAs are sufficient to trigger the type I interferon signaling in mouse lungs, which can robustly counteract the rapid activation of ILC2 cells upon exposure to IL-33 or a fungal allergen. Our current data on the protective function of exRNA derived from RV1B and AdV5 infected cell seem to be contradictory to the clinical experience where respiratory viral infections frequently cause exacerbations in asthmatic patients, who often have defective interferon responses [26–28]. However, until now, the exact molecular mechanism for virus-induced asthma exacerbations remain unexplained mechanistically. Therefore, more works are needed to in order to fully

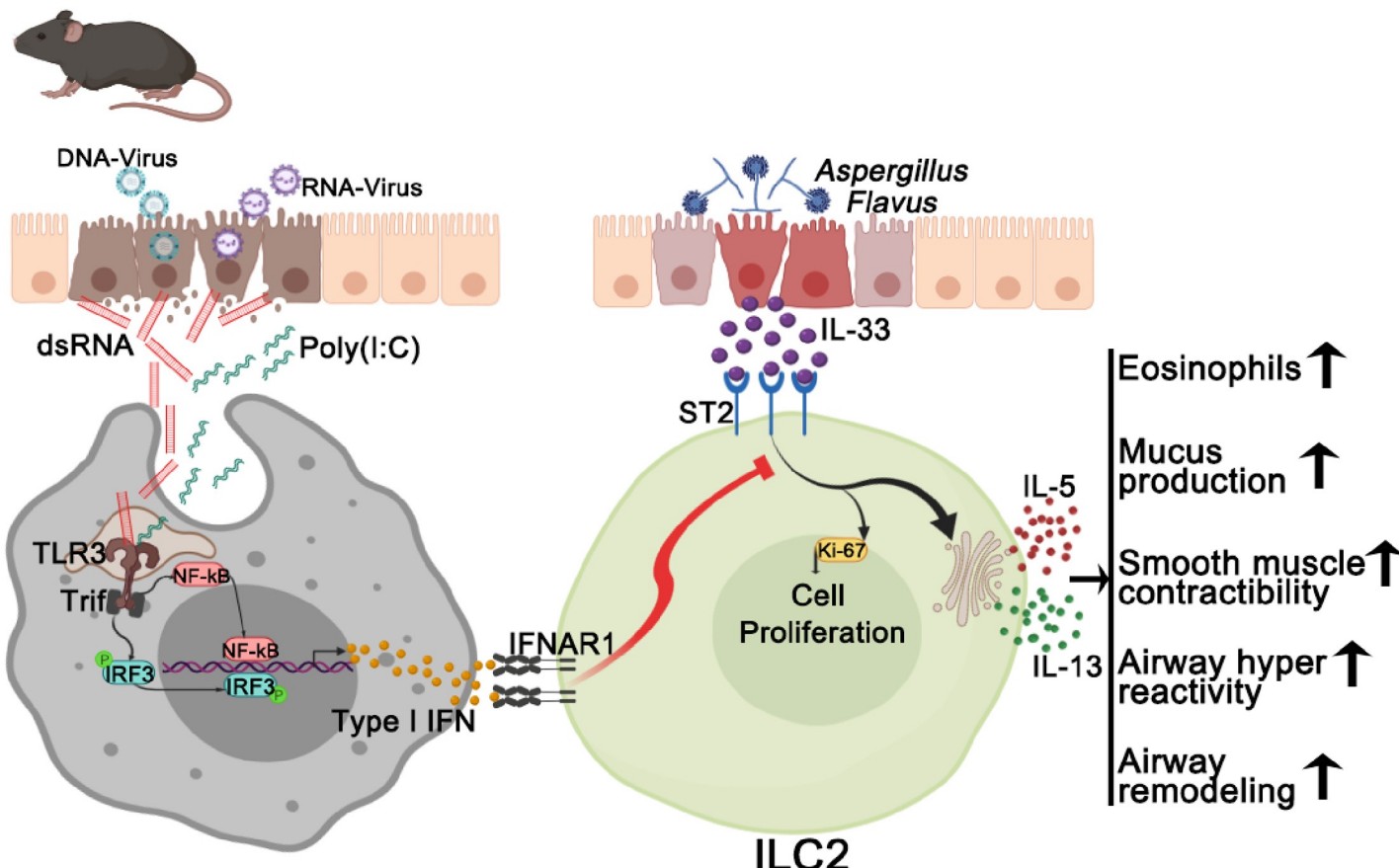

**Fig 10. A proposed working model illustrates the potential mechanism through which exRNAs inhibit ILC2-driven acute type 2 lung inflammation (Artwork was initially created in BioRender, https://app.biorender.com).**

address underlying molecular mechanisms of the effector function of exRNAs-induced innate immune signaling on ILC2-mediated type 2 inflammation.

Here, we report that immune sensing and signaling of exRNA seem to play a regulatory role in controlling the development of type 2 lung immunopathology induced by IL-33 or a fungal allergen. Specifically, the protective effects of exRNA were mediated by local interferon responses that suppressed the function and proliferation of ILC2 cells. Our results may provide novel insights into the mechanisms underlying susceptibility to a wide range of allergic disorders, including asthma, rhinosinusitis and food allergy.

## Supporting information

**S1 Fig. FACS gating strategy for the identification of specific cell types including eosinophils.** Cells recovered from BALF were stained for cell surface markers as indicated. The absolute numbers were calculated based on reference beads (top panel). Formula (Number of cells per mL): (Total Beads/# events of Beads) x (# event of Sample)/Volume of Tested Sample. (Related to Figs 1C, 2C, 3C, 5A, 6A, 7A, 8A, 9A and **S3A**).
(PPTX)

**S2 Fig. FACS gating strategy for the identification of lung ILC2 cells.** Cells recovered from total lung homogenates were stained for cell surface markers as indicated. The absolute numbers were calculated based on reference beads (top panel). Formula (Number of cells per mL): (Total Beads/# events of Beads) x (# event of Sample)/Volume of Tested Sample. (Related to Figs 1E, **4**B–**4**D, 5C, 6C, 7C, 8C, 9C and **S5C**).
(PPTX)

**S3 Fig. Poly(I:C) inhibits IL-33-induced eosinophilia in Rag1$^{-/-}$ mice. A.** Four groups of Rag1$^{-/-}$ mice were treated with PBS, Poly(I:C), IL-33 or IL-33+Poly(I:C) as indicated. BALF was collected and analyzed for differential immune cell types. **B.** Administration of Poly(I:C) decreased the percentage and number of eosinophils in lungs after exposure to IL-33. (n = 2–5 per group as indicated with open circles, P value was determined using Mann-Whitney test, $^{*}$ p < 0.05, $^{**}$ p < 0.01, $^{***}$ p < 0.001).
(PPTX)

**S4 Fig. Cells infected with both RNA- and DNA viruses produce immunostimulatory dsRNA species. . A.** Detection of the double-stranded structure in total RNA isolated from virus-infected HeLa cells by a dsRNA-specific antibody J2 with Dot Blot. **B.** Induction of IFNs by RNAs produced by HeLa cells infected with both RNA- and DNA-viruses. BEAS-2B cells were transfected with total RNAs (1.0 µg /ml per 0.2x10$^6$ cells) of HeLa cells infected with viruses for 16–18 hours. Total RNAs were also treated with or without RNases III or T1 as indicated. **C.** Transcriptional induction of gene expressions by RV1B RNA in mouse lungs. Wild type mice were exposed via the intratracheal route to increased amounts of RV1B RNA as indicated. After 16-18h, total RNA isolated from mouse lungs was subjected to RT-qPCR analysis. (Related to Figs 8 & 9). N.D., not detected.
(PPTX)

**S5 Fig. Administration of RV1B RNA into TLR3$^{-/-}$ partially inhibited *A. flavus*-induced lung inflammation. A.** Groups of WT and TLR3$^{-/-}$ mice as indicated were treated with *A. flavus* or *A. flavus*+RV1B RNA. BALF was collected and analyzed for differential immune cell types. **B.** The percentage and number of eosinophils cells in lungs of WT and TLR3$^{-/-}$ mice were analyzed. **C.** The percentage and number of ILC2 cells and percentage of IL5$^+$/IL13$^+$-double positive ILC2 cells in lungs of WT and TLR3$^{-/-}$ mice were analyzed. (n = 5–6 per group

as indicated with open circles, P value was determined using Mann-Whitney test, P value ≥0.05 was not considered statistically significant [N.S.]). * p < 0.05, ** p < 0.01). **D.** Cells were stained with the isotype antibodies corresponding to IL-5 and IL-13.
(PPTX)

## Acknowledgments

We thank Dr. Nu Zhang for critical reading of the manuscript and Ms. Karla Gorena for technical assistance in flow cytometry. We are grateful to Dr. Nu Zhang for the help with FACS analysis.

## Author Contributions

**Conceptualization:** Li She, Hamad H. Alanazi, Xiao-Dong Li.

**Data curation:** Li She, Hamad H. Alanazi, Xiao-Dong Li.

**Formal analysis:** Li She, Hamad H. Alanazi, Yong Liu, Xin Zhang, Xiao-Dong Li.

**Funding acquisition:** Xiao-Dong Li.

**Investigation:** Li She, Hamad H. Alanazi, Liping Yan, Yilun Sun, Xiao-Dong Li.

**Methodology:** Li She, Liping Yan, Edward G. Brooks, Peter H. Dube, Fushun Zhang, Yilun Sun, Yong Liu, Xin Zhang, Xiao-Dong Li.

**Project administration:** Xiao-Dong Li.

**Resources:** Edward G. Brooks, Peter H. Dube, Yan Xiang, Fushun Zhang, Yilun Sun, Yong Liu, Xin Zhang, Xiao-Dong Li.

**Software:** Xiao-Dong Li.

**Supervision:** Edward G. Brooks, Peter H. Dube, Xiao-Dong Li.

**Validation:** Li She, Hamad H. Alanazi, Liping Yan, Yilun Sun, Xiao-Dong Li.

**Visualization:** Xiao-Dong Li.

**Writing – original draft:** Li She, Xiao-Dong Li.

**Writing – review & editing:** Li She, Hamad H. Alanazi, Liping Yan, Edward G. Brooks, Peter H. Dube, Yan Xiang, Yong Liu, Xin Zhang, Xiao-Dong Li.

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
