## [Decision Letter · Decision Letter 0]

30 Apr 2020

PONE-D-20-07673

Sensing and Signaling of Immunogenic Extracellular RNAs Restrain Group 2 Innate Lymphoid Cell-Driven Acute Lung Inflammation and Airway Hyperresponsiveness

PLOS ONE

Dear Dr. Li,

Thank you for submitting your manuscript to PLOS ONE. After careful consideration, we feel that it has merit but does not fully meet PLOS ONE’s publication criteria as it currently stands. Therefore, we invite you to submit a revised version of the manuscript that addresses the points raised during the review process.

Although the manuscript is very interesting, there are a number of points that the reviewers, who are experts in the field, have raised and which have to be addressed. First of all, you have to give more details on the methodology as well as on the experiments performed in the M&M section. Second, the statistical analysis has to be revised. Third, the comments of reviewer #2 regarding gating strategy and intracellular staining have to be addressed. Finally, the conclusions drawn have to clearly reflect what the experimental design allows to infer from the experiments: e.g. a poly(I:C) only control appears to be missing, poly(I:C) was given together with allergen and/or IL-33 and not alone, the use of TLR3 single knockouts is recommended. Please provide a detailed point-by-point response addressing all the points raised by the reviewers along with your revised manuscript.

We would appreciate receiving your revised manuscript by Jun 14 2020 11:59PM. To enhance the reproducibility of your results, we recommend that if applicable you deposit your laboratory protocols in protocols.io, where a protocol can be assigned its own identifier (DOI) such that it can be cited independently in the future. For instructions see: http://journals.plos.org/plosone/s/submission-guidelines#loc-laboratory-protocols

We look forward to receiving your revised manuscript.

Kind regards,

Heinz Fehrenbach

Academic Editor

PLOS ONE

Journal Requirements:

1.

2. At this time, we request that you please report additional details in your Methods section regarding animal care, as per our editorial guidelines: 1) Please provide details of animal welfare (e.g., shelter, food, water, environmental enrichment) 2) please describe any steps taken to minimize animal suffering and distress, such as by administering analgesics, 3) please include the method of sacrifice and 4) Please describe the post-operative care received by the animals, including the frequency of monitoring and the criteria used to assess animal health and well-being. Thank you for your attention to these requests.

3. Please provide additional information about each of the cell lines used in this work, including source, history, culture conditions and any quality control testing procedures (authentication, characterisation, and mycoplasma testing). For more information, please see http://journals.plos.org/plosone/s/submission-guidelines#loc-cell-lines.

4. Please provide the source, product number and any lot numbers of the reagents listed in the "Cells, viruses and reagents" section of the Materials and Methods.

5. At this time, we ask that you please provide scale bars on all IHC images.

Reviewers' comments:

Reviewer's Responses to Questions

**Comments to the Author**

1. Is the manuscript technically sound, and do the data support the conclusions?

Reviewer #1: Partly

Reviewer #2: Yes

2. Has the statistical analysis been performed appropriately and rigorously? 

Reviewer #1: No

Reviewer #2: Yes

3. Have the authors made all data underlying the findings in their manuscript fully available?

Reviewer #1: Yes

Reviewer #2: No

4. Is the manuscript presented in an intelligible fashion and written in standard English?

Reviewer #1: Yes

Reviewer #2: Yes

5. Review Comments to the Author

Reviewer #1: This study aimed to investigate the effect of extracellular RNA or DNA on the activation of innate lymphoid Type II cells. Therefore the authors used a mouse model for experimental asthma triggered with a fungal antigen. Applying the extracellular RNA suppressed the product the rotation of the ILC2 cells which was shown to be dependent on the TLR3-Trif-IFNAR1 axis.

This is an interesting and elegantly performed study, however in my opinion there are several limitations which are further explained in the comments below, especially concerning the statistical analysis.

Major points

• The mouse model needs to be described in more detail in the methods part. How many challenges were performed, at which time points and was there also a control receiving poly IC only but no allergen/IL33. Please also provide a scheme in the figure for the fungal model.

• The statistical analysis is performed only with a parametric t test. But actually due to the in vivo experiments a non-parametric tests might be more appropriate. Second since there are more groups and treatments being compared at the same time this analysis certainly requires a more advanced statistical tests.

• For all figures containing histology pictures it would be good to supply an overview picture off the whole lung lobe to judge the overall inflammation.

• In figure 1B the authors show a dose increase of poly IC, but with different doses than were used in a mouse model. However , they conclude from this figure that 5 microgram would be the ideal dose for further experiments. This is confusing.

• The text mentions the DCS and T cells are strongly increased at 50 microgram of poly IC, but there seems to be a nonsignificant trend only. This would need to be adapted in the results.

• It is interesting that poly IC reduces the ILC2 and th2 response in the lungs. Did the authors check for i.e. a th1 or an ILC 1 response as viral products might reduce the th2 phenotype by augmenting the th1 response?

• One major limitation of this paper is that the IL 33 or allergen was given at the same time as the poly IC. In order to investigate if such a stimulus is protected or aggravates asthmatic inflammation one would need to first establish experimental asthma in the mouse and then do the challenge with poly IC.

Minor points

• In my opinion that different knockouts different models but with the same analysis could be shown in the supplement to reduce the amount of figures.

Reviewer #2: The study looks at the effect of co-treatment with IL-33 and TLR3 agonists on lung inflammation, using a number of mouse knockouts to make a convincing case for an IFN pathway downstream of poly IC or dsRNA treatment.

There are a couple of points which could be resolved in order to improve the manuscript.

In places the description of the methods is not detailed enough.

The time and number of doses is not always clear. The degree of IL-33 induced inflammation in the lung is very high for what appears to be 3 treatments with a fairly standard dose.

How were ILC2 numbers calculated. The ILC2 plots in the figures do not appear to match the gating strategy described.

For intracellular cytokine staining, the authors should provide isotype controls in cells from both unstimulated mice. This is critical give the staining increase in small, and it is possible that the ILC2 change in size or in terms of intrinsic fluorescence.

In the final experiments, I'm not convinced about the use of TLR3/MAVS double knockouts. While MAVS will be important in viral infection, it is less clear if it would be involved in response to extracellular DNA, which is what the authors are examining. This should be repeated in TLR3 single knockouts.

More details should be given regarding the animals (age / sex / diet / light cycle (including time of IL-33 treatment) /randomisation ect)

6. PLOS authors have the option to publish the peer review history of their article (what does this mean?). If published, this will include your full peer review and any attached files.

Reviewer #1: No

Reviewer #2: No

---

## [Author Response · Author response to Decision Letter 0]

8 Jun 2020

please also see the attached rebuttal letter.

PONE-D-20-07673

“Sensing and Signaling of Immunogenic Extracellular RNAs Restrain Group 2 Innate Lymphoid Cell-Driven Acute Lung Inflammation and Airway Hyperresponsiveness”

We thank the editor and both reviewers for the constructive comments. We have performed the requested experiments. The generated new data have been included in this revised manuscript. In addition, we have also added more details on the sections of Materials and Methods as suggested by the reviewers. The detailed changes in text have been highlighted in yellow. Our point-to-point responses are shown below.

Editor’s comments:

“you have to give more details on the methodology as well as on the experiments performed in the M&M section.”

Re: yes, we have added more details on the relevant sections of Materials and Methods. The changes are highlighted in yellow.

“the statistical analysis has to be revised.” 

Re: yes, we have revised the statistical analysis as recommended by Reviewer #1 and added more details to the corresponding figure legends and the relevant sections. 

“the comments of reviewer #2 regarding gating strategy and intracellular staining have to be addressed.” 

Re: yes, we have provided more details on the gating strategy and intracellular staining (see our responses to the reviewer #2’s concerns below)

“the conclusions drawn have to clearly reflect what the experimental design allows to infer from the experiments: e.g. a poly(I:C) only control appears to be missing, poly(I:C) was given together with allergen and/or IL-33 and not alone, the use of TLR3 single knockouts is recommended.”

Re: yes, the poly (I:C) alone control has been included as shown in Figures 2-6. As suggested, we have performed a new experiment using TLR3 single knockouts. The generated result is now shown in Figure S5. Compared to DKO mice (TLR3-/-MAVS-/-), in which the inhibitory effects of RV1B RNA were completely abolished, TLR3 single knockouts only showed a partial phenotype, indicating that both TLR3 and MAVS are both involved in sensing RV1B RNA.

Reviewers' comments:

Reviewer #1: 

“This study aimed to investigate the effect of extracellular RNA or DNA on the activation of innate lymphoid Type II cells. Therefore, the authors used a mouse model for experimental asthma triggered with a fungal antigen. Applying the extracellular RNA suppressed the product the rotation of the ILC2 cells which was shown to be dependent on the TLR3-Trif-IFNAR1 axis. This is an interesting and elegantly performed study, however in my opinion there are several limitations which are further explained in the comments below, especially concerning the statistical analysis.”

Re: we really appreciate the Reviewer’s positive comments on this work.

Major points 

“The mouse model needs to be described in more detail in the methods part. How many challenges were performed, at which time points and was there also a control receiving poly IC only but no allergen/IL33. Please also provide a scheme in the figure for the fungal model.”

Re: yes, the poly (I:C) alone control has been included as shown in Figures 2-6. As shown in the scheme, poly (I:C), RV1B RNA or AdV5 RNA, was administered on Day -1, 1 and 2. Yes, we have provided a scheme in the figure for the fungal model as shown in Figures 3A, 4A, 8A and 9A.

“The statistical analysis is performed only with a parametric t test. But actually, due to the in vivo experiments a non-parametric tests might be more appropriate. Second since there are more groups and treatments being compared at the same time this analysis certainly requires a more advanced statistical tests.”

Re: we agree with the reviewer’s comment and have reanalyzed the relevant data set using the Mann-Whitney test. Accordingly, we have updated the P values in the related figures. The changes on the statistical analysis did not affect our overall conclusions.

“For all figures containing histology pictures it would be good to supply an overview picture off the whole lung lobe to judge the overall inflammation.”

Re: we thank the reviewer’s suggestion. As the overall inflammation of lung had been reflected on the BALF via FACS analysis, we feel that the current the magnification (scale bars, 100�m) of lung histology could provide a better resolution on a more detailed view of the airway via showing the infiltration of leukocytes, mucus production and goblet-cell hyperplasia. 

“In figure 1B the authors show a dose increase of poly IC, but with different doses than were used in a mouse model. However, they conclude from this figure that 5 microgram would be the ideal dose for further experiments. This is confusing.”

Re: the decision to use 5 �g was not solely based on Figure 1B. In fact, all experiments shown in Figure 1 have contributed to the adoption of 5 �g per mouse as the optimal dose for further extensive experiments.

“The text mentions the DCS and T cells are strongly increased at 50 microgram of poly IC, but there seems to be a nonsignificant trend only. This would need to be adapted in the results.”

Re: we agree with the reviewer’s comments. The relevant texts have been removed. 

“It is interesting that poly IC reduces the ILC2 and th2 response in the lungs. Did the authors check for i.e. a th1 or an ILC 1 response as viral products might reduce the th2 phenotype by augmenting the th1 response?”

Re: we thank the reviewer’s excellent point. We agree that it would be very interesting to examine a Th1 or ILC 1 response in the future experiments.

“One major limitation of this paper is that the IL 33 or allergen was given at the same time as the poly IC. In order to investigate if such a stimulus is protected or aggravates asthmatic inflammation one would need to first establish experimental asthma in the mouse and then do the challenge with poly IC.”

Re: we agree with the reviewer that we have only shown the preventative role of poly(I:C). It would be important to examine whether or not the poly IC administration could attenuate the ongoing eosinophilic lung inflammation induced by IL-33 or a fungal allergen in mice in the future experiments. 

Minor points

“In my opinion that different knockouts different models but with the same analysis could be shown in the supplement to reduce the amount of figures.”

Re: we appreciate the reviewer’s suggestion. However, at the moment, we think that the story flowed well with the current order of figures. We are willing to consider move some figures to the supplement, if there is a publication limit on the number of main figures.

Reviewer #2: 

“In places the description of the methods is not detailed enough.”

Re: yes, we have added more details on the relevant sections of Materials and Methods. The changes are highlighted in yellow.

“The time and number of doses is not always clear. The degree of IL-33 induced inflammation in the lung is very high for what appears to be 3 treatments with a fairly standard dose.”

Re: we did 3 treatments with either IL-33 or A. flavus to induce lung inflammation. We have provided an experimental scheme in the relevant figures.

“How were ILC2 numbers calculated. The ILC2 plots in the figures do not appear to match the gating strategy described.”

Re: We apologize for the confusion of ILC2 plot. The ILC2 gating strategy shown in Figure 4B was used for calculating the absolute number of ILC2. The gating plots shown in Figure 4 C-D were the representative the showing the percentage of ILC2 cells in total lung lymphocytes. Now, we have moved the old Figure 2B into supplementary figures to make it as the new Figure S2. Same as the method for calculating the number of eosinophils, ILC2 numbers were calculated based on the reference beads shown in the new Figure S2. The formula (Number of ILC2 cells per mL) is (total beads/# events of beads) x (# event of sample)/ Volume of cell per mL). 

“For intracellular cytokine staining, the authors should provide isotype controls in cells from both unstimulated mice. This is critical give the staining increase in small, and it is possible that the ILC2 change in size or in terms of intrinsic fluorescence.”

Re: we thank the reviewer’s excellent suggestion. For the intracellular cytokine staining, now we have added isotype control antibodies corresponding to IL5- or IL13- staining, respectively (Figure S5D).

“In the final experiments, I'm not convinced about the use of TLR3/MAVS double knockouts. While MAVS will be important in viral infection, it is less clear if it would be involved in response to extracellular DNA, which is what the authors are examining. This should be repeated in TLR3 single knockouts.”

Re: we thank the reviewer’s excellent point. As suggested, we have performed new experiments using TLR3 single knockouts. The generated new data is now shown in Figure S4. 

“More details should be given regarding the animals (age / sex / diet / light cycle (including time of IL-33 treatment) /randomisation etc).”

Re: yes, we have added more details on animals in the relevant section of Materials and Methods.

---

## [Decision Letter · Decision Letter 1]

14 Jul 2020

Sensing and Signaling of Immunogenic Extracellular RNAs Restrain Group 2 Innate Lymphoid Cell-Driven Acute Lung Inflammation and Airway Hyperresponsiveness

PONE-D-20-07673R1

Dear Dr. Li,

We’re pleased to inform you that your manuscript has been judged scientifically suitable for publication and will be formally accepted for publication once it meets all outstanding technical requirements.

Kind regards,

Heinz Fehrenbach

Academic Editor

PLOS ONE

Additional Editor Comments (optional):

Reviewers' comments:

Reviewer's Responses to Questions

**Comments to the Author**

1. If the authors have adequately addressed your comments raised in a previous round of review and you feel that this manuscript is now acceptable for publication, you may indicate that here to bypass the “Comments to the Author” section, enter your conflict of interest statement in the “Confidential to Editor” section, and submit your "Accept" recommendation.

Reviewer #1: All comments have been addressed

Reviewer #2: All comments have been addressed

2. Is the manuscript technically sound, and do the data support the conclusions?

Reviewer #1: Yes

Reviewer #2: Yes

3. Has the statistical analysis been performed appropriately and rigorously? 

Reviewer #1: Yes

Reviewer #2: Yes

4. Have the authors made all data underlying the findings in their manuscript fully available?

Reviewer #1: Yes

Reviewer #2: Yes

5. Is the manuscript presented in an intelligible fashion and written in standard English?

Reviewer #1: Yes

Reviewer #2: Yes

6. Review Comments to the Author

Reviewer #1: the authors have answered all of my comments and have they are with improved the manuscript. I do not have further points.

Reviewer #2: (No Response)

7. PLOS authors have the option to publish the peer review history of their article (what does this mean?). If published, this will include your full peer review and any attached files.

Reviewer #1: No

Reviewer #2: No

---

## [Editor Report · Acceptance letter]

17 Jul 2020

PONE-D-20-07673R1 

Sensing and Signaling of Immunogenic Extracellular RNAs Restrain Group 2 Innate Lymphoid Cell-Driven Acute Lung Inflammation and Airway Hyperresponsiveness 

Dear Dr. Li:

I'm pleased to inform you that your manuscript has been deemed suitable for publication in PLOS ONE. Congratulations! Your manuscript is now with our production department. 

Kind regards, 

on behalf of

Prof. Dr. Heinz Fehrenbach 

Academic Editor

PLOS ONE